# Nitrate/Ammonium Ratios and Nitrogen Deficiency Impact on Nutrient Absorption and Photosynthetic Efficiency of *Cedrela odorata*

Sulianne Idalior Paião Rosado [1], José Zilton Lopes Santos [1], Ives San Diego Amaral Saraiva [1], Nonato Junior Ribeiro dos Santos [1], Tainah Manuela Benlolo Barbosa [1] and Josinaldo Lopes Araujo [2,*]

[1] College of Agricultural Sciences, Federal University of Amazonas, 6200, General Rodrigo Octavio Jordão Ramos Ave, Manaus 69080-900, Brazil; sulianneidalior@gmail.com (S.I.P.R.); ziltton@yahoo.com.br (J.Z.L.S.); sevi_san22@hotmail.com (I.S.D.A.S.); nonatojr.rs@gmail.com (N.J.R.d.S.); tainahbenlolo@yahoo.com.br (T.M.B.B.)

[2] Department of Agricultural Sciences, Federal University of Campina Grande, Campina Grande 58840-000, Brazil

* Correspondence: josinaldo.lopes@professor.ufcg.edu.br; Tel.: +55-84-99694-6067

**Abstract:** Nitrate ($NO_3^-$) and ammonium ($NH_4^+$) are the primary forms of nitrogen (N) taken up by plants and can exhibit different effects on plant nutrition, photosynthesis, and growth. The objective was to investigate the influence of nitrate/ammonium proportions (%) on the nutritional status, photosynthetic parameters, and the development of *Cedrela odorata* seedlings after 150 days of cultivation. We tested six nitrate/ammonium ratios (100/0; 80/20; 60/40; 40/60; 20/80; and 0/100 of $NO_3^-$ and $NH_4^+$, respectively), plus a control treatment (without N supply). Based on the results, the species responds to the supply of N; however, the $NO_3^-$ and $NH_4^+$ proportions did not show any significant effect on plant growth. The deficiency of nitrogen (N) in *Cedrela odorata* decreases the photosynthetic rate, nutrient absorption, and initial growth of this species. Increasing the proportion of N in the form of nitrate inhibited the absorption of S (sulfur) but did not interfere with the accumulation of N, Ca (calcium), Mg (magnesium), Mn (manganese), Zn (zinc), B (boron), and Cu (copper). *Cedrela odorata* apparently does not distinguish between nitrate and ammonium in the N absorption process, since the proportions between these forms of N did not affect its photosynthetic rate, nutrient accumulation, or growth.

**Keywords:** ammonium toxicity; chlorophyll index; gas exchange; tree seedling nutrition; mineral nitrogen; nitrogen forms; red cedar

## 1. Introduction

*Cedrela odorata* is considered noble for the national and international market for sawn wood [1,2] and has excellent characteristics to substitute forest species with high wood potential, such as mahogany [3]. Although the native tropical forests adapt to the high acidity and low natural fertility of the soils [4,5], N fertilization can positively affect the seedling growth of these species [6–8]. However, the efficiency of nitrogen fertilization varies with the proportions of nitrate ($NO_3^-$) and ammonium ($NH_4^+$) in the growth medium, as well as the ability of species or genotypes to take advantage of these forms of nitrogen [9,10].

Nitrogen (N) is an essential constituent of proteins, nucleotides, and nucleic acids such as DNA and RNA [9,11]. The low N availability also results in a lower photosynthetic rate due to the breakdown of the thylakoid membranes, affecting the capture of light energy and, consequently, carbon fixation and plant growth and development [12,13]. In addition, there is a reduction in the chlorophyll content and fluorescence values [12], causing chlorosis, initially on older leaves [14].

Plants preferentially absorb $NO_3^-$ and $NH_4^+$ [15]. The proportion between the two forms in the growth substrate is a factor that can influence the process of assimilation of N by plants, with consequences for their development and growth [16,17]. Plants adapted to soils with high pH often tend to absorb $NO_3^-$ preferentially, whereas those from environments with low pH prefer $NH_4^+$ [18,19]. Species from the Meliaceae family, such as *C. odorata*, occur in acidic soils [2,20]. In an acidic environment, $NH_4^+$ is the N form most abundant due to acidity inhibiting the activity of nitrifying microorganisms that cause oxidation of free $NH_4^+$ [18]. It indicates that C. *odorata* can have a preference for $NH_4^+$ or a combination of $NO_3^-$ with $NH_4^+$, where $NH_4^+$ predominates.

In general, fertilizers containing only $NH_4^+$ limit plant growth and development, and few species perform well when $NH_4^+$ is the only form of N provided [20,21]. It is due to the high levels of free $NH_4^+$ producing changes in cell pH and ionic and hormonal imbalances [20,22]. In addition, $NH_4^+$ can cause a reduction in photosynthetic speed and biomass production (root, stem, and leaves) as a consequence of leaf chlorosis and necrosis and the decline of the root system and stem [10,23].

Conversely, $NO_3^-$ can be stored by plants at high levels and translocated from among the tissues, often without deleterious effects [24]. In general, plants supplied with $NO_3^-$ produce greater leaf area and growth compared to plants supplied with $NH_4^+$ [25,26]. However, Araujo et al. [16] reported that some species, such as rice, do not have the capacity to assimilate nitrate in the initial growth phase. Thus, some researchers recommend using an 80:20 (%) ratio of $NO_3^-$ and $NH_4^+$, respectively, primarily when it is not known the species preference [16,27], but this proportion may not be suitable for some species [17].

The efficiencies of N absorption and use by plants are dependent on both the predominant N form in the substrate and the species preference [10,20,28]. Furthermore, ammonium competes with cations while nitrate competes with anions in the ionic absorption process, which can cause a reduction in the absorption of mineral nutrients in plants, depending on the availability of these forms of N in the growing medium [22,25].

Thus, it was investigated the hypothesis that there is an adequate proportion of $NO_3^-$ $NH_4^+$ that promotes a better nutritional and photosynthetic status and development of C. odorata seedlings.

## 2. Materials and Methods

### 2.1. Species and Growth Conditions

*C. odorata* belongs to the Meliaceae family and is found especially in the Central Amazon region, where soils with high acidity and low fertility levels predominate, such as ultisols and oxisols rich in clay [2,29,30]. These soils have limitations such as low natural fertility, high acidity, and organic matter content ranging from 2 to 3.1 g $kg^{-1}$ [31]. The region's climate is tropical, classified as Aw, according to the classification of Peel et al. [32].

The present study was conducted using *C. odorata* seedlings at 60 days old. These were grown in a greenhouse with 612.3 $\mu$mol $m^{-2}$ $s^{-1}$ of photosynthetic photon flux density, photoperiod 12:00 h/12:00 (day/night), an average temperature of 35.7 °C, and a relative humidity of 63% in Manaus, Amazonas State, Brazil.

### 2.2. Treatments and Experimental Design

The treatments consisted of six different proportions of $NO_3^-$/$NH_4^+$ (100/0; 80/20; 60/40; 40/60; 20/80; and 0/100, respectively), in a constant supply of 200 mg $dm^{-3}$ of N, plus a control treatment (without N fertilization: 0/0). The plant pots were distributed in a randomized complete block design, with five replications and one plant per experimental unit, totaling 35 plants.

### 2.3. Specific Procedures

2.3.1. Soil Collection, Characterization, and Fertilization

Soil samples of the Xanthic Haplustox [33] collected in the subsurface (20–40 cm depth) from the secondary rainforest in Manaus, Amazonas State, Brazil, were used. Then, the

soil was air-dried, homogenized, and passed through a 4 mm sieve to remove large debris and stones. Before the experiment, soil sub-samples were sieved (2 mm) and analyzed to determine their physical and chemical properties.

The soil sample presented 74% clay (pipette method) and showed: pH in water 4.1; 1.42 g kg$^{-1}$ of soil organic matter; 9.1mg K kg$^{-1}$; 0.1 cmol$_c$ calcium (Ca) kg$^{-1}$; 1.3 cmol$_c$ magnesium (Mg) kg$^{-1}$; 2.0 mg phosphorus (P) kg$^{-1}$ (resin extractant); 6.7 mg sulfur (S) kg$^{-1}$; 0.1 cmol$_c$ aluminum (Al) kg$^{-1}$; 103.0 mg iron (Fe) kg$^{-1}$; 0.3 mg manganese (Mn) kg$^{-1}$; 0.2 mg zinc (Zn) kg$^{-1}$; 0.1 mg boron (B) kg$^{-1}$; and 0.2 mg copper (Cu) kg$^{-1}$ [34].

Soil samples (aggregate size of 4 mm) received lime at the equivalent rate of 2.3 g dm$^{-3}$ of a mixture of $CaCO_3$ and $MgCO_3$. $7H_2O$ in a stoichiometric ratio of Ca:Mg of 4:1, aiming to increase the base saturation to 60%. Then, the combination (lime plus the entire soil mass) was incubated for 30 days.

After incubation, the soil was fertilized [35,36] at the following rates: 100 mg of K; 455 mg of P; 40 mg of S; 3.6 mg of Mn; 4 mg of Zn; 0.8 mg of B; 1.3 of Cu; and 0.15 mg of Mo per dm$^{-3}$. The sources used were, respectively, KCl; $KH_2PO_4$; $Ca(H_2PO_4)$ $2H_2O$; $H_3PO_4$; $CaSO_4$; $S°$; $MnSO_4$ $H_2O$; $ZnSO_4.7H_2O$; $H_3BO_3$; and $CuSO_4.5H_2O$. In the same period, the N treatments were applied as $Ca(NO_3)_2.4H_2O$ and $(NH_4)_2SO_4$.

The liming and phosphorus fertilization were applied in solid form (powder). At the same time, all other nutrients and treatments were supplied in the form of a solution, using pure reagents for analysis (p.a.). The sources were mixed with the entire soil mass before potting 4 dm$^3$ individually in each polyethylene pot and again incubating for 30 days. Then, one soil sample was collected from each pot, and the soil was mixed into one composite sample to make a composite sample for each treatment. The pH in water was 4.9; 1.52 g kg$^{-1}$ of soil organic matter; 123.5 mg K kg$^{-1}$; 5.34 cmol$_c$ Ca kg$^{-1}$; 1.12 cmol$_c$ Mg kg$^{-1}$; 396.64 mg P kg$^{-1}$ (resin extractant); 249.97 mg S kg$^{-1}$; 0.1 cmol$_c$ Al kg$^{-1}$; 162.0 mg Fe kg$^{-1}$; 7.95 mg Mn kg$^{-1}$; 8.61 mg Zn kg$^{-1}$; 0.08 mg B kg$^{-1}$; and 3.04 mg Cu kg$^{-1}$. Although the ammonium sulfate source has an acidifying potential [37], its influence on soil pH was not observed.

### 2.3.2. Seedlings, Containers, Irrigation, and Topdressing Fertilization

*Cedrela odorata* L. seeds were germinated in a plastic tray (60 × 40 × 10 cm) of length, width, and height, respectively, filled with vermiculite. Both seeds and substrate were sterilized in sodium hypochlorite 0.3% (*v/v*). Deionized water was applied daily to meet plant requirements. At 60 days after germination (DAG), seedlings of the single stem and two to four pairs of leaves were uniformly selected based on height and transplanted to polyethylene pots with controlled drainage pores of 4 dm$^3$ (34.00 × 7.63 × 15.60 cm) of height, lower diameter, and upper diameter, respectively.

During the soil incubation periods and plant growth, the soil moisture was maintained at 60% of field capacity by the daily weighing of pots and the addition of deionized water, according to the principles of [38]. Supplemental K at 50 mg dm$^{-3}$ of K was applied to the 90th and 120th days after the transplant (DAT). Potassium monobasic phosphate ($KH_2PO_4$) was dissolved in water and applied in solution form using a graduated pipette on the surface of each pot.

### 2.4. Experiment Evaluation

At 150 days after transplantation, we evaluated the treatment effects on plants using measures of changes in photosynthesis, growth, and nutritional parameters.

### 2.5. Photosynthetic Parameters
#### 2.5.1. Gas Exchange

Measured the net photosynthetic rate (*A*), stomatal conductance (*gs*), and transpiration rate (*E*) in young leaves when wholly expanded. The evaluations occurred between 07:00 and 11:00 a.m. using an infrared gas analyzer (IRGA–LI-COR 6400, Biosciences Inc., Nebraska, EUA) [39]. The measurements were made with a photon flow density (PPFD) of

1000 µmol m$^{-2}$ s$^{-1}$, an initial $CO_2$ concentration of $380 \pm 4$ µmol mol$^{-1}$, a leaf temperature of $31 \pm 1$ °C, and water vapor around $21 \pm 1$ mmol mol$^{-1}$.

### 2.5.2. Chlorophyll Fluorescence

The measurement was carried out on the third pair of leaflets from the apex of the plant in the morning hours (10:00–11:00 h). Before measurement, the leaves of red cedar selected for analysis were adapted to darkness for 30 min using special plastic clips in order to initiate all reaction centers to open and minimize physiological processes associated with the energization of the thylakoid membrane. The chlorophyll *a* fluorescence was measured by a portable fluorometer (PEA, MK2, 9600, Hansatech, Norfolk, UK). After the adaptation of leaflets to darkness, the adaxial leaf side was then exposed to a 5 s excitation pulse of high irradiance (3000 µmol m$^{-2}$ s$^{-1}$) with a wavelength of 650 nm. Then, the minimal ($F_o$), maximum ($F_m$), and variable fluorescence ($F_v$) were determined. The measured data were analyzed by the JIP test [40,41] and used to calculate the maximal quantum yield of PSII photochemistry ($F_v/F_m$) and the response of photosystem II (PSII) by means of the performance index on an absorption basis ($PI_{ABS}$) and the total performance index on an absorption basis ($PI_{total}$).

### 2.5.3. Relative Chlorophyll Index (RCI)

The RCI was evaluated on the same leaves used to assess liquid photosynthesis and gas exchange using a hand-held chlorophyll meter (SPAD-502, Konika Minolta, Osaka, Japan). The RCI readings were taken in the midpoint of the second pair of leaflets of each leaf, in the space between the main vein, from 10:00 a.m. to 2:00 p.m. at two replications. Therefore, the RCI value corresponds to the mean of two readings obtained from each experimental unit.

### *2.6. Growth Parameters*

### 2.6.1. Plant Height and Diameter

The plant height was measured from the soil surface to the top of the plant using a graduated ruler (cm), and the stem diameter was measured 1.0 cm above the soil surface using a digital pachymeter (mm) (Mitutoyo-500, Mitutoyo, Miyazaki, Japan). At this time, the leaf growth was observed through the leaf area (cm$^2$) using the CI-202 portable area meter to take the measurements.

### 2.6.2. Dry Matter Production

The plants were harvested 0.5 cm above the soil surface using a garden scissor, and separated into leaves, stems, petioles, and roots. The roots were carefully separated from the soil using a 2 mm mesh sieve and tap water. After that, all the parts were washed with tap water to remove any attached particles, rinsed twice with distilled water, and dried at 65 °C to obtain a constant weight ($\approx$72 h). Later, the shoot dry matter (SDM) (SDM = leaves + stems and petioles) and root dry matter (RDM) were determined, and total dry matter (TDM; TDW = SDM + RDM), using a digital scale (accurate to 0.001 g). Finally, the root-to-shoot ratio (RSR) was calculated using the following equation: RSR = RDM/SDM.

### *2.7. Nutritional Parameters*

### Nutritional Status

Dried leaves, obtained as described in Section 2.6.2, were ground in a Willey mill and sieved through a 1 mm screen and subsequently stored at room temperature in hermetically sealed plastic bottles with a capacity of 50 mL [42]. A sample of 0.5 g, previously dried in an oven at 65 °C, was accurately weighed and digested using 5 mL of a 2:1 (*v/v*) mixture of $HNO_3$ and HCl ratio in macro digestion tubes (80 mL) on a block digester at 210 °C [42]. The P was determined by visible spectrophotometry ($\lambda$ = 680 nm) (spectrophotometer Micronal® model B-580); potassium (K) was determined by flame photometer Micronal®

model B-462; and calcium (Ca); magnesium (Mg); iron (Fe); manganese (Mn); zinc (Zn); and copper (Cu) were determined by atomic absorption spectroscopy, AAS (GBC, model Avanta Sigma, Braeside, Australia).

For N determination, samples of 0.1 g were digested in 2 mL of $H_2SO_4$ + 0.5 mL of $H_2O_2$ on a plate heater at 200 °C, the digested solution was made alkaline with excess NaOH solution, and the ammonia was distilled using a semi-micro-Kjeldahl apparatus (Marconi, MA036, Brazil) and titrated using 0.02 mol $L^{-1}$ $H_2SO_4$ [42]. Total S was extracted with $BaCl_2$ in 1 mL of an acid solution (HCl 6 mol $L^{-1}$ + 20 mg $L^{-1}$ of S) and determined by the turbidimetric method ($\lambda$ = 420 nm). The B was determined by the curcumin colorimetric method ($\lambda$ = 540 nm) [42]. Nutrient accumulations (AN) in leaves were calculated as follows: AN (mg plant$^{-1}$) = foliar nutrient concentration (mg kg$^{-1}$) × leaf dry matter (kg). Complementarily, the symptoms of nutritional deficiency were characterized and described according to their appearance.

### 2.8. Data Analysis

All the data were subjected to outlier detection and checked for normality (Shapiro–Wilk) and variance homogeneity (Levene) at a 0.05 level of significance. Then, the results were statistically analyzed using one-way ANOVA ($p < 0.1$ and $< 0.05$ significance levels), unless otherwise indicated. When the F test was significant, the treatment means were compared using Tukey's test.

### 3. Results

### 3.1. Foliar Nutrient Contents and Accumulation

The different proportions of $NO_3^-$/$NH_4^+$ influenced significantly ($p < 0.05$, Appendix A) the leaf contents of N, Ca, Mg, S, Mn, Zn, B, and Cu (Table 1) but did not affect the leaf concentrations of K, P, and Fe. Because there is no reference for leaf nutrient concentrations in *C. odorata* plants, adequate levels were used in plant tissues for comparison [6] and control treatment data [43].

**Table 1.** Effect of $NO_3^-$/$NH_4^+$ ratios on the foliar contents of nitrogen (N), phosphorus (P), potassium (K), calcium (Ca), magnesium (Mg), sulfur (S), manganese (Mn), zinc (Zn), boron (B), copper (Cu), and iron (Fe) in plants of Cedrela odorata at 150 days after transplantation.

| $NH_4^+$/$NO_3^-$ (%) | N (g kg$^{-1}$) | P (g kg$^{-1}$) | K (g kg$^{-1}$) | Ca (g kg$^{-1}$) | Mg (g kg$^{-1}$) | S (g kg$^{-1}$) |
|---|---|---|---|---|---|---|
| 100/0 | 41.00 ± 1.00 a | 3.00 ± 0.00 abc | 20.00 ± 1.41 a | 15.00 ± 2.00 ab | 3.00 ± 0.50 a | 4.00 ± 0.60 e |
| 80/20 | 41.0.0 ± 1.41 a | 3.60 ± 0.55 a | 22.20 ± 2.68 a | 15.40 ± 2.07 ab | 2.80 ± 0.45 ab | 5.40 ± 0.55 cd |
| 60/40 | 39.40 ± 1.95 ab | 2.20 ± 0.84 bc | 21.20 ± 1.30 a | 9.80 ± 5.60 c | 1.00 ± 0.71 c | 5.00 ± 1.00 de |
| 40/60 | 39.00 ± 2.12 ab | 3.40 ± 0.89 ab | 23.20 ± 2.39 a | 16.80 ± 0.84 a | 3.00 ± 0.10 a | 7.00 ± 1.00 b |
| 20/80 | 38.20 ± 0.84 b | 2.00 ± 0.71 c | 21.80 ± 1.30 a | 12.20 ± 2.49 bc | 2.00 ± 0.10 b | 7.00 ± 2.00 b |
| 0/100 | 37.40 ± 0.89 b | 4.00 ± 0.71 a | 23.00 ± 1.22 a | 16.60 ± 1.40 a | 2.80 ± 0.84 ab | 6.40 ± 0.55 cd |
| 0/0 | 20.40 ± 0.55 c | 2.20 ± 0.45 bc | 21.00 ± 1.41 a | 18.80 ± 0.45 a | 3.0 0 ± 0.10 a | 14.00 ± 1.00 a |

| - | Mn (mg kg$^{-1}$) | Zn (mg kg$^{-1}$) | B (mg kg$^{-1}$) | Cu (mg kg$^{-1}$) | Fe (mg kg$^{-1}$) | |
|---|---|---|---|---|---|---|
| 100/0 | 99.0 ± 7.4 b | 45.0 ± 1.2 a | 45.0 ± 2.0 b | 2.0 ± 0.0 b | 177.0 ± 5.5 a | |
| 80/20 | 127.0 ± 2 3.0 ab | 46.0 ± 3.4 b | 52.0 ± 3.0 b | 2.4 ± 0.9 b | 142.4 ± 37.4 ab | |
| 60/40 | 115.6 ± 20.7 ab | 61.2 ± 11 b | 41.4 ± 7.8 b | 4.8 ± 2.5 a | 138.4 ± 12.9 ab | |
| 40/60 | 116.6 ± 1.81 ab | 44.0 ± 1.6 b | 47.2 ± 0.8 b | 4.0 ± 1.0 ab | 143.2 ± 22.6 ab | |
| 20/80 | 132.0 ± 17.8 a | 39.8 ± 1.1 b | 47.0 ± 0.7 b | 3.4 ± 0.5 ab | 142.8 ± 27.2 ab | |
| 0/100 | 119.0 ± 25.8 ab | 41.6 ± 9.4b | 46.8 ± 4.3 b | 3.2 ± 0.8 ab | 142.6 ± 21.9 ab | |
| 0/0 | 82.2 ± 14.3 | 30.0 ± 2.1 c | 97.6 ± 0.6 a | 2.8 ± 0.4 ab | 123.8 ± 26.5 b | |

The values presented are the mean ± standard error ($n$ = 5). Different lowercase letters after the number indicate significant differences among treatments by the Tukey's test ($p < 0.05$).

In general, the treatment with a higher proportion of N-$NO_3^-$ showed the highest absolute value of leaf N content (41.05 g kg$^{-1}$) (Table 1); however, it did not differ significantly from the other ratios of $NO_3^-$/$NH_4^+$. The proportions of 60/40, 20/80, and 0/100 of

$NO_3^-/NH_4^+$ showed foliar N contents of 39.5, 38.18, and 37.6 g kg$^{-1}$, respectively. These plants exhibited mild symptoms of toxicity as a light green color in the leaf blades of old leaves, and these started 103 days after the treatment application. In contrast, the plants supplied with the 40/60 ratio of $NO_3^-/NH_4^+$ presented greater growth of the aerial part, and the leaves were well expanded. Although the leaf N concentrations found in the 40:60 ratio (38.93 g kg$^{-1}$ of N) are slightly higher than those observed in the 20:80 proportions of $NO_3^-/NH_4^+$ (38.18 g kg$^{-1}$) and 0/100 (37.60 g kg$^{-1}$), all these values are above the range considered adequate for suitable plant growth [43].

The symptoms of ammonium toxicity manifest as low development of the root system, foliar chlorosis, and necrosis [44], besides the overall suppression of plant growth [45]. In the present study, it was observed that there was a shortening of the root system and leaf chlorosis and necrosis 150 days after transplantation. The more intense symptoms were associated with levels of ammonium up to 120 mg dm$^{-3}$.

In plants of the control treatment, it was observed a lower concentration of N (20.45 g kg$^{-1}$) and lower foliar contents of Mn (Table 1) and Zn (Table 1). Concerning N content, this value was below the level considered adequate for plant tissues (32.2 g kg$^{-1}$) [43]. These plants showed deficiency symptoms 80 days after transplantation, such as chlorosis of old leaves (Figure 1), a decrease in leaf area, the lower release of new leaves, and a reduction in plant growth. This response corroborates the N content found in plant leaves.

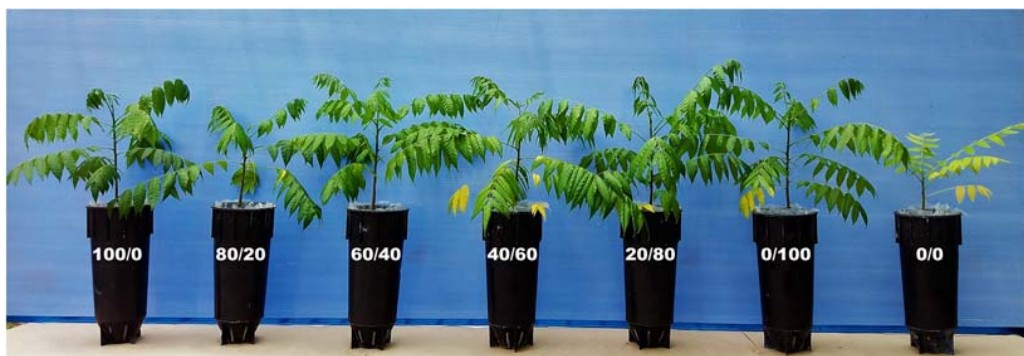

**Figure 1.** Visual aspects of *Cedrela odorata* growing under different $NO_3^-/NH_4^+$ ratios 150 days after treatments.

In the other treatments, the foliar N concentrations were between 36.4 and 44.0 g kg$^{-1}$, remaining above the levels considered adequate (32.2 g kg$^{-1}$) for good plant growth [43]. About foliar S concentration, the value varied between 3.72 and 13.96 g kg$^{-1}$. The lower amount was associated with the treatments with a greater proportion of N-$NO_3^-$ (Table 1). Still, these lower values of foliar S content were above the level (2.74 g kg$^{-1}$) considered to be adequate for plant growth [43].

In general, the control treatment presented leaf contents of 93.45 and 13.96 for Ca and S, respectively, and 488.83 mg kg$^{-1}$ for B. The concentrations of Ca, S [43], and B [6] were above the range considered adequate for plant tissues. Furthermore, it is interesting to note that the 60:40 ratio of $NO_3^-/NH_4^+$ promoted the lowest values of leaf contents (49.81 g kg$^{-1}$, 4.88 g kg$^{-1}$, 5.22 g kg$^{-1}$, and 206.8 mg kg$^{-1}$ of Ca, Mg, S, and B, respectively). However, all values were above the levels considered adequate for the appropriate development of plants [6,43].

The treatments significantly influenced the leaf accumulation of N, K, P, Fe, Mn, and Zn ($p < 0.05$, Appendix A) and Ca, Mg, S, and Cu ($p < 0.01$) in the *C. odorata* plants (Figure 2).

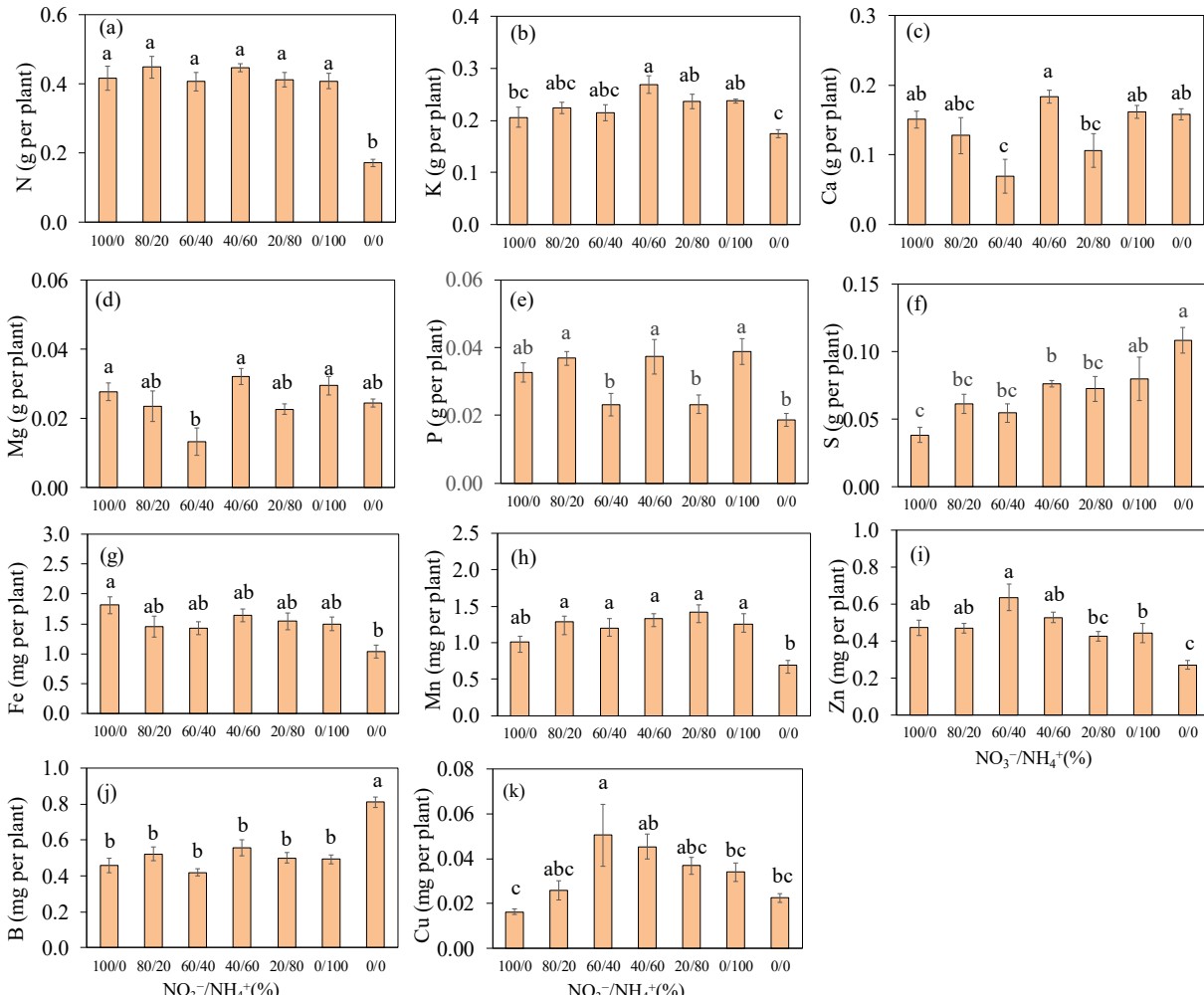

**Figure 2.** Effect of $NO_3^-/NH_4^+$ ratios on the foliar accumulation of N (**a**), K (**b**), Ca (**c**), Mg (**d**), P (**e**), S (**f**), Fe (**g**), Mn (**h**), Zn (**i**), B (**j**), and Cu (**k**) in plants of *Cedrela odorata* at 150 days after treatments. The error bars represent the standard error of the mean. Bars followed by the same letters are not significantly different at $p < 0.05$ (variables **c,d,f,k**) and $p < 0.1$ (variables **a,b,e,g,h,i**), according to a one-way ANOVA followed by Tukey's test.

Concerning only the effects of the $NO_3^-/NH_4^+$ ratio, these did not have any statistically significant impact on the foliar accumulation of N, K, Fe, Mn, Zn, and B (Figure 2a,b,g–j); however, they affected the uptake of Ca, Mg, P, S, and Cu (Figure 2c–f,k). The effect of $NO_3^-/NH_4^+$ proportions depended on the type of element, with the 40:60 ratio of $NO_3^-/NH_4^+$ providing the highest values of Ca, Mg, and P accumulation (Figure 2c–e). In contrast, the 0/100 and 60/40 ratios of $NO_3^-/NH_4^+$ presented the highest values of foliar accumulation of S (Figure 2f) and Cu (Figure 2k). In general, the S uptake by plants increased almost linearly with the reduction in nitrate concentration and an increase in the amount of ammonium in the soil (Figure 2f).

Conversely, the 60/40 ratio of $NO_3^-/NH_4^+$ and control treatment contributed to the lowest values of foliar nutrient accumulation (Figure 2). Overall, the pattern of foliar nutrient contents and accumulation was similar.

### 3.2. Photosynthetic Parameters

The treatments significantly affected the parameters $A$ ($p < 0.1$), $gs$ ($p < 0.05$), $E$ ($p < 0.05$), $F_v/F_m$ ($p < 0.05$), $PI_{ABS}$ ($p < 0.1$), $PI_{total}$ ($p < 0.1$), and IRC ($p < 0.05$) (Figure 3). However, the $NO_3^-/NH_4^+$ proportions did not significantly influence the variables $PI_{ABS}$, $PI_{total}$, and IRC (Figure 3).

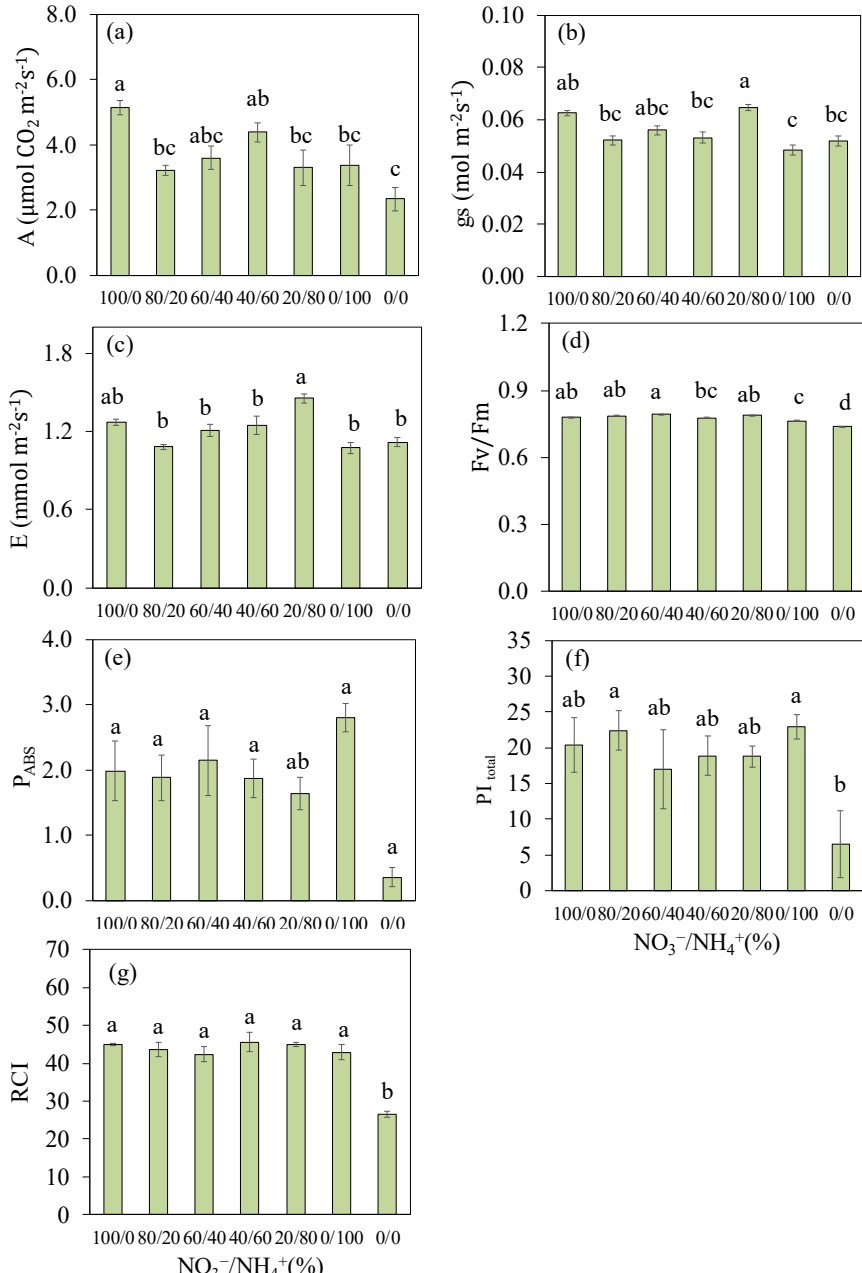

**Figure 3.** Effect of $NO_3^-/NH_4^+$ ratios on net photosynthesis rate (A): (**a**) stomatal conductance (*gs*) and (**b**) Table 150 days after treatments. The error bars represent the standard error of the mean. Bars followed by the same letters are not significantly different at $p < 0.05$ (variables **b**–**g**) and $p < 0.1$ (variable **a**) according to the one-way ANOVA followed by Tukey's test.

The 100/0 and 40/60 ratios of $NO_3^-/NH_4^+$ presented the highest values of *A* (Figure 3a), while the 100/0 and 20/80 proportions of $NO_3^-/NH_4^+$ showed the highest levels of *gs*, *E*, and Fv/Fm (Figure 3b–d). Concerning the Fv/Fm variable, these ratios did not significantly differ in the 80/20 and 60/40 proportions of $NO_3^-/NH_4^+$ (Figure 3d). The parameters *A*, gs, *E*, and IRC varied between 3.25 and 5.15; 0.04 and 0.06; 1.08 and 1.45; and 42.9 and 45.52, respectively, between the different $NO_3^-/NH_4^+$ proportions.

In general, there was a decrease in the values of photosynthetic parameters when the N was provided only as ammonium and the mean for the variables *A*, *gs*, and $F_v/F_m$ (Figure 3a,b,d). Concerning the Fv/Fm variable, although their values varied as a function of $NO_3^-/NH_4^+$ ratio, they stayed in the range (0.73 to 0.79) considered suitable for forest

species [46,47]. In the case of the performance index on the absorption basis, the 0/100 ratio of $NO_3^-/NH_4^+$ (Figure 3a) reduced the levels of both $PI_{ABS}$ and $PI_{total}$ (Figure 3e,f).

Similarly, to that observed for the nutritional variables, the control treatment reduced the value of all the variables except for the *gs* parameter (Figure 3). The absence of N presented a reduction of 11.5, 87.1, and 71.6% in Fv/Fm, $PI_{ABS}$, and $PI_{total}$, respectively, compared to the treatments with the highest values, 60/40, 0/100, and 0/100 ratios of $NO_3^-/NH_4^+$ (Figure 3d–f).

### 3.3. Plant Growth

The treatments did not significantly affect ($p < 0.1$) the growth parameters of height and diameter (Figure, 4a,b), SDM, RDM, and leaf area ($p < 0.1$) (Figure 4c–f). There was only a considerable difference between the proportions of $NO_3^-/NH_4^+$ and the control treatment.

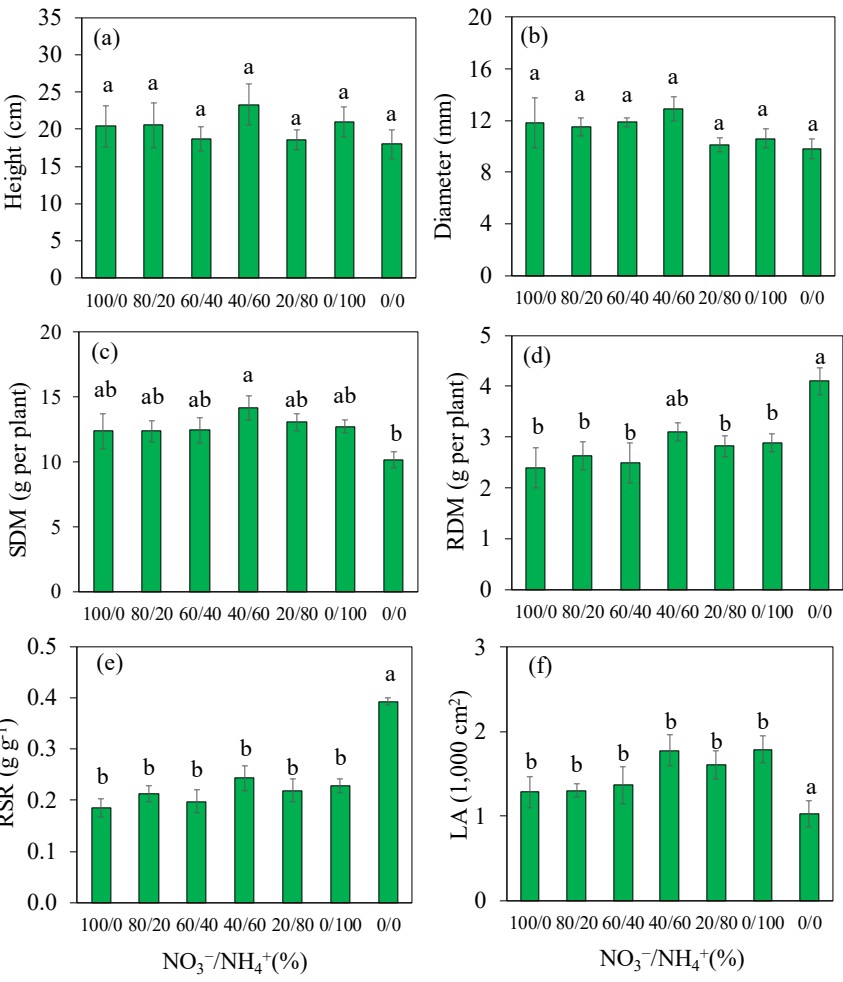

**Figure 4.** Effect of $NO_3^-/NH_4^+$ ratios on height (**a**); diameter (**b**); shoot dry matter (SDM) (**c**); root dry matter (RDM) (**d**); root-to-shoot ratio (RSR) (**e**); and leaf area (LA) (**f**); in the Cedrela odorata plant at 150 days after treatments. The error bars represent the standard error of the mean. Bars followed by the same letters are not significantly different at $p < 0.1$, according to a one-way ANOVA followed by Tukey's test.

Regarding the response of the plants to the N forms, despite no significant difference among $NO_3^-/NH_4^+$ proportions, there was a slight tendency for the 40/60 ratio of $NO_3^-/NH_4^+$ to present a higher height, diameter, SDM, RDM, and LA value (Figure 4a–d,f).

It is interesting to note that the control treatment showed the lowest values of both SDM (Figure 4c) and leaf area (Figure 4f). On the other hand, this treatment promoted the

highest levels of the MSR (Figure 4d) and SRS (Figure 4e) variables. However, in the case of RDM, the control treatment did not significantly differ from the 40:60 ratio of $NO_3^-/NH_4^+$.

## 4. Discussion

This present study focused on a better understanding of the effect of N forms and the impact of deficiency of this element on the early development of *C. odorata*. The $NO_3^-/NH_4^+$ ratio had little effect on the initial development of the plants. However, the absence of N causes a substantial restriction on the development of this species. Although the treatments slightly influenced the nutrient concentrations in the leaves of the plants, these contents were within the range considered adequate [6–8,43] for proper growth, regardless of the $NO_3^-/NH_4^+$ proportions in the soil.

Higher availability of N-$NO_3^-$ in the soil can result in increased absorption of K, Mg, and Ca by plants, supposedly by ionic competition [48]. This premise does not apply to the plants of *C. odorata* since the plants showed higher yield values in treatments with the joint supply of $NO_3^-/NH_4^+$, with a predominance of $NH_4^+$, even though the forms did not result in a significant difference. This effect may be related to species adaptation and preference [18–20,48].

The deficiency symptoms observed in plants grown without N (0 mg $dm^{-3}$ of N) highlight the great importance of this nutrient to the species and indicate that supply below the plant's metabolic demand promotes visual and morphological changes in the plants [9,11]. N is a constituent of proteins and nucleic acids, and its deficiency affects the metabolism as it reduces cell division and expansion. It is a consequence of the alteration at the molecular level, which leads to cellular change, which, finally, results in tissue disorganization and the appearance of visual symptoms, common to all species [12,14] as well as having been documented in other studies for forest species [49,50].

The response of forest species to N forms is contrasting since some species adapt to ammonium, nitrate, or both. It may be due to nutritional requirements or the availability of N in the environment [20,45]. In the present study, the N forms caused little effect on nutrient content, mainly in N, K, Fe, Mn, Zn, and B accumulation. For *Eucalyptus urophylla*, proportions of N-$NO_3^-$/N-$NH_4^+$ (0/100, 25/75, 50/50, 75/25, and 100/0) had little impact on the nutritional status and photosynthetic parameters of the plants after 70 days of growth [51].

Concerning the effect of N-$NO_3^-$ on the reduction in S uptake, it is probably because S uptake occurs through an active process and may be reduced in the presence of some anions due to competition in absorptive sites [22,52]. In addition, the S-$SO_4^{2-}$ needs to be reduced to sulfide ($S^{2-}$) before being assimilated into cysteine [53]. This process competes with $NO_3^-$ reduction in metabolic energy as ATP [54,55].

Conversely, the $NH_4^+$ tends to limit the absorption of K, Mg, and Ca [22,25]; however, we have not observed this effect in this work. However, in the proportions of 60/40, 20/80, and 0/100 (%) of $NO_3^-/NH_4^+$, there is slight chlorosis in the older leaves, probably because $NH_4^+$ reduces the synthesis of organic acids [20]. The symptoms of $NH_4^+$ toxicity vary widely and are species-dependent [10,56]. In general, some species that occur in acidic or low-redox soil conditions tend to be tolerant of $NH_4^+$ [45].

In general, the use of $NO_3^-$ and $NH_4^+$ had a much better effect when supplied together. The supply of mixed $NO_3^-/NH_4^+$ from the same source eliminates the disadvantages of each isolated form, maintains cellular and environmental pH stability, improves the absorption of nutrients, and maintains an adequate proportion of N assimilation [20,25]. In addition, the presence of $NO_3^-$ can reduce the toxic effect of $NH_4^+$ and enhance the growth of plants by inducing the synthesis of enzymes to produce cytokinins; this effect depends on the $NO_3^-/NH_4^+$ proportion and species [17].

Regarding the effects of N forms on the growth of *C. odorata*, the joint supply of $NO_3^-$ and $NH_4^+$ with a predominance of $NH_4^+$ tended to enhance the height, diameter, and production of SDM and increase in LA in *C. odorata* plants, however, without showing any significant difference (Figure 4a–c,f). These results indicate that for the studied species, the

supply of $NH_4^+$ probably did not result in decreased cell expansion, osmotic regulation, carbon accumulation, or growth [12,17].

The plasticity of *C. odorata* plants to N forms is also highlighted as a function of the little effect of the proportions of $NO_3^-/NH_4^+$ on photosynthetic variables (*A*, *E*, $F_v/F_m$, $PI_{ABS}$, $PI_{total,}$ and IRC). As the forms of N did not affect the foliar concentrations of N, the plants met the metabolic requirements related to photosynthetic parameters. However, Tsabarducas et al. [57] mentioned that PSII photoinhibition is probably not responsible for the differential sensitivity of the N forms.

Thus, the lack of a pattern of response in the photosynthetic variables in *C. odorata* seedlings in the function of one form of N proportional to another reinforces the hypothesis that there is no clear preference of the species for some type of N. This effect can be a mechanism of adaptation to the availability of N in the environment [10,20]. Plants that prefer $NH_4^+$ are generally adapted to acidic soils or those with low redox potential, where possibly nitrification is weak [20,21]. As *C. odorata* occurs in acidic soils and sometimes in soils with low-redox potential, it was expected that this species would prefer ammonium. However, this preference was not observed in this study. Chlorophyll fluorescence of *Olea europaea* in controlled conditions showed a lack of plant response to $NO_3^-$, $NH_4^+$, and urea N forms [57]. Conversely, *Pinus radiata* seedlings at 150 days after transplanting show higher rates of photosynthesis associated with treatments where nitrate (80%) predominates compared to ammonium [58].

Although the N-$NO_3^-$/N-$NH_4^+$ ratios did not influence the development of the plants in the present study, this species seems to be highly demanding for N. This premise is evident because when the plants were grown in the absence of N, they showed the lowest values of nutrient contents and nutrient accumulation, decreasing in production of SDM and LA (reduction of 28.5 and 42.5%, respectively) compared to the treatment with the highest values of these variables.

In addition, there was a significant reduction in photosynthetic variables *A*, *E*, $F_v/F_m$, $PI_{ABS}$, $PI_{total,}$ and RCI, which are related to the use of assimilates by the plant, where the N deficiency stimulates higher proportions of carbon for the formation of starch [23,59]. If this accumulation in the chloroplast is in excess, photosynthesis can be affected by preventing the arrival of $CO_2$ at Rubisco's carboxylation sites [23]. Thus, there is a breakdown in the thylakoid membranes, impairing the capture of light energy, reducing photosynthetic rates, and causing stress to the plant. In general, the scarcity of N affects the structure and functioning of the photosynthetic apparatus [13,60,61] and $PI_{ABS}$, which reflects the functioning process of the photosynthetic apparatus [61]. Nitrogen fertilization of *C. fissilis* growing under controlled conditions for 210 days enhanced the accumulation of nutrients, production of dry matter, leaf area, and RCI due to the increase in nitrogen fertilization [6].

In the present study, the N content in leaves corroborates closely with the values of RCI, where foliar concentrations below the range considered adequate for the plants (control treatment) provided a lower chlorophyll index. The RCI positively correlates with the chlorophyll content, being considered a suitable indicator to evaluate the effects of N on the plant and the nutritional status of the leaves [62,63]. This occurs because around 50 to 70% of the total N in leaves is part of enzymes associated with chloroplasts, where chlorophyll is located [63,64].

The *C. odorata* plants with N deficiency directed the metabolic resources toward the growth of the root system. It can be a strategy to improve N capture in the substrate as an adaptation mechanism; however, this is implied by the loss of metabolic reserves in shoots [20,65]. Consequently, the more significant root growth of this species contributed to higher RDM and RSR values.

## 5. Conclusions

*Cedrela odorata* plants require nitrogen (N) in the initial phase of cultivation when cultivated in soil with low natural fertility and low organic matter content. The N deficiency in *C. odorata* decreases the photosynthetic rate, nutrient absorption, and initial growth of

this species. Increasing the proportion of N in the form of nitrate inhibited the absorption of S (sulfur) but did not interfere with the accumulation of N, Ca (calcium), Mg (magnesium), Mn (manganese), Zn (zinc), B (boron), and Cu (copper). *C. odorata* does not distinguish between nitrate and ammonium in the N absorption process, since the proportions between these forms of N did not affect its photosynthetic rate, nutrient accumulation, or growth. *C. odorata* plants require N in the initial phase of growth when cultivated in soil with low natural fertility and low organic matter content.

**Author Contributions:** Conceptualization, J.Z.L.S. and S.I.P.R.; methodology, J.Z.L.S.; software, I.S.D.A.S.; validation, N.J.R.d.S.; formal analysis, J.Z.L.S.; investigation, S.I.P.R.; resources, J.Z.L.S. and S.I.P.R.; data curation, J.Z.L.S. and S.I.P.R.; writing—original draft preparation, J.Z.L.S. and S.I.P.R.; writing—review and editing, J.L.A.; visualization, T.M.B.B.; supervision, J.Z.L.S.; project administration, J.Z.L.S.; funding acquisition, J.Z.L.S. All authors have read and agreed to the published version of the manuscript.

**Funding:** This research received no external funding.

**Institutional Review Board Statement:** Not applicable.

**Data Availability Statement:** The data used to support the findings of this study are included within the article.

**Acknowledgments:** Acknowledgments are due to the Postgraduate Program in Tropical Agriculture of the Federal University of Amazonas for infrastructure research.

**Conflicts of Interest:** The authors declare no conflict of interest.

**Appendix A**

**Table A1.** Summary of analysis of variance for variables related to nutrient contents and accumulation, photosynthesis, and plant growth.

| Source of Variance | DF | N | P | K | Ca | Mg | S | Mn |
|---|---|---|---|---|---|---|---|---|
| | | **Mean Square** | | | | | | |
| **Treatment** | **6** | **264.96 \*\*** | **3.12 \*\*** | 6.49 ns | 46.05 \*\* | 2.86 \*\* | 54.09 \*\* | 1466.45 \*\* |
| Error | 28 | 1.72 | 0.42 | 3.11 | 3.63 | 0.20 | 0.37 | 361.07 |
| CV (%) | - | 3.59 | 22.46 | 8.11 | 12.75 | 17.79 | 8.74 | 16.80 |
| | | **Zn** | **B** | **Cu** | **Fe** | **N-ac** | **P-ac** | **K-ac** |
| Treatment | 6 | 433.51 \*\* | 1909.71 \*\* | 4.56 \* | 1278.59 ns | 0.047 \*\* | 0.000 \*\* | 0.004 \*\* |
| Error | 28 | 31.74 | 28.78 | 1.31 | 575.71 | 0.002 | 0.000 | 0.001 |
| CV (%) | - | 12.82 | 9.96 | 35.51 | 16.63 | 13.84 | 24.44 | 13.68 |
| | | **Ca-ac** | **Mg-ac** | **S-ac** | **Mn-ac** | **Zn-ac** | **B-ac** | **Cu-ac** |
| Treatment | 6 | 0.007 \*\* | 0.000 \*\* | 0.0024 \*\* | 0.304 \*\* | 0.061 \*\* | 0.082 \*\* | 0.0007 \*\* |
| Error | 28 | 0.001 | 0.000 | 0.0004 | 0.051 | 0.0088 | 0.005 | 0.0001 |
| CV (%) | - | 29.22 | 25.94 | 28.77 | 19.36 | 20.32 | 13.95 | 42.54 |
| | | **Fe-ac** | *A* | *gs* | *E* | **Fv/Fm** | **P$_{ABS}$** | **PItotal** |
| Treatment | 6 | 0.285 \* | 4.05 \*\* | 0.000 \*\* | 0.09 \*\* | 0.000 \*\* | 2.72 \*\* | 152.60 ns |
| Error | 28 | 0.083 | 0.77 | 0.000 | 0.00 | 0.000 | 0.59 | 62.49 |
| CV (%) | - | 19.44 | 24.29 | 6.81 | 7.91 | 0.59 | 42.48 | 43.68 |
| | | **RCI** | **Height** | **Diameter** | **SDM** | **RDM** | **RSR** | **LA** |
| Treatment | 6 | 224.60 \*\* | 13.94 ns | 6.07 ns | 7.28 ns | 1.64 \*\* | 0.02 \*\* | 399,849.35 \* |
| Error | 28 | 13.63 | 25.98 | 0.72 | 3.86 | 0.40 | 0.001 | 141,211.00 |
| CV (%) | - | 8.88 | 25.18 | 19.34 | 15.80 | 21.76 | 17.13 | 25.92 |

CV: coefficient of variation. \*\* $p < 0.01$ and \* $p < 0.05$ by the F test. The word ns stands for the non-significant data at a 5% probability level. Degrees of freedom (DF). Nutrient symbols followed by *ac* (for example, N-ac) denote the accumulation of the respective nutrient in the dry mass of the aerial part. *A*: net photosynthesis rate. *gs*: stomatal conductance. *E*: transpiration rate. $F_v/F_m$: maximal quantum yield of PSII photochemistry ($F_v/F_m$). PI$_{ABS}$: performance index on an absorption basis. PI$_{total}$: performance index on an absorption basis. RCI: relative chlorophyll index. SDM: shoot dry matter. RDM: root dry matter. RSR: root-to-shoot ratio. LA: leaf area.

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
