# Peer review of "Nitrate/Ammonium Ratios and Nitrogen Deficiency Impact on Nutrient Absorption and Photosynthetic Efficiency of Cedrela odorata"

_nitrogen, doi:10.3390/nitrogen5010001_

Round 1
Reviewer 1 Report
Comments and Suggestions for Authors
The study titled “Nitrate/ammonium ratios and nitrogen lack impacts nutrients absorption and photosynthetic efficiency of Cedrela odorata” has been reviewed. In my opinion, the study is very interesting and the presentation is fair. It is in particular a good work as it addressed a fundamental global issue of food security via the Organic Agriculture route. Specifically, it underscores the importance of nitrogen as a vital element in plant’ growth and wellbeing and also went on to show the negative impact of its deficiency in plant. This work will contribute greatly to the attainment of the SDGs 1 and 2 when published. However, there are critical issues that must be addressed before the manuscript can be fit for acceptance. Authors should succinctly address the following issues:
Title
The current title “Nitrate/ammonium ratios and nitrogen lack impacts nutrients absorption and photosynthetic efficiency of Cedrela odorata” is good but shows that something is missing. I suggest that authors should recap it. More importantly, I suggest changing the “lack” to “deficiency”.
Abstract
The section is very informative and contains all the basics of the manuscript content. The following issues needs to be addressed nonetheless
1. The level of English usage is poor and needs a total overhaul
2. Abbreviations should not be used except it has first been fully written before subsequent abbreviation e.g., Nitrogen should be written in full in the first line
3. Authors should avoid the use of personal pronouns e.g., “we” which is very rampart here
Keywords
Authors should increase the numbers of keywords for this manuscript and focus on words and phrases that best describe the overall content of the manuscript and that are useful for abstracting and indexation
Introduction
1. The level of English usage is poor and needs a total overhaul
2. Authors should avoid the use of personal pronouns e.g., “we” which is very rampart here
Materials and Methods
1. The statement “C. odorata is belonging to the Meliaceae family, and this originates in the central Amazon region, where predominate clayey Argisols and Oxisols, as well-drained [2,29,30]” seem to be incorrectly written. This need to be rechecked
2. Authors should avoid the use of personal pronouns especially “we” which is overwhelmingly used in this section
Figures
Two different figures are labelled as figure 2 and there is no figure 1 at all.
References
There are too many old references in this section which can be changed if possible. This is a very current and trending topic and published works of the last 5 years should form the basis of the references used.
General Comments
1. There is need the authors to really underscore the mechanism behind Cedrela odorata not distinguishing between nitrate and ammonium in the N absorption process because this study shows that the proportions between these forms of N did not affect its photosynthetic rate, nutrient accumulation or growth.
2. I suggest that authors do a thorough language editing of the manuscript
3. I discourage authors from using personal pronouns throughout the manuscript
Comments on the Quality of English Language
There is need for English editing especially the use of personal pronouns
Author Response
REVIEWER 1
Comments and Suggestions for Authors
The study titled “Nitrate/ammonium ratios and nitrogen lack impacts nutrients absorption and photosynthetic efficiency of Cedrela odorata” has been reviewed. In my opinion, the study is very interesting and the presentation is fair. It is in particular a good work as it addressed a fundamental global issue of food security via the Organic Agriculture route. Specifically, it underscores the importance of nitrogen as a vital element in plant’ growth and wellbeing and also went on to show the negative impact of its deficiency in plant. This work will contribute greatly to the attainment of the SDGs 1 and 2 when published. However, there are critical issues that must be addressed before the manuscript can be fit for acceptance. Authors should succinctly address the following issues:
Title
The current title “Nitrate/ammonium ratios and nitrogen lack impacts nutrients absorption and photosynthetic efficiency of Cedrela odorata” is good but shows that something is missing. I suggest that authors should recap it. More importantly, I suggest changing the “lack” to “deficiency”.
Response: The authors partially agree with the reviewer, changing “lack” to “deficiency”. However, we had difficulty making further changes to the title without making it too long.
Abstract
The section is very informative and contains all the basics of the manuscript content. The following issues needs to be addressed nonetheless
- The level of English usage is poor and needs a total overhaul.
Response: Corrected
- Abbreviations should not be used except it has first been fully written before subsequent abbreviation e.g., Nitrogen should be written in full in the first line
Response: Corrected
- Authors should avoid the use of personal pronouns e.g., “we” which is very rampart here
Response: Corrected
Keywords
Authors should increase the numbers of keywords for this manuscript and focus on words and phrases that best describe the overall content of the manuscript and that are useful for abstracting and indexation
Response: Corrected. Others keywords were added
Introduction
- The level of English usage is poor and needs a total overhaul.
Response: Corrected in whole manuscript
- Authors should avoid the use of personal pronouns e.g., “we” which is very rampart here
Response: Corrected in whole manuscript
Materials and Methods
- The statement “C. odorata is belonging to the Meliaceae family, and this originates in the central Amazon region, where predominate clayey Argisols and Oxisols, as well-drained [2,29,30]” seem to be incorrectly written. This need to be rechecked
Response: This part of the text has been modified
- Authors should avoid the use of personal pronouns especially “we” which is overwhelmingly used in this section
Response: Corrected in whole manuscript
Figures
Two different figures are labelled as figure 2 and there is no figure 1 at all.
Response: Corrected
References
There are too many old references in this section which can be changed if possible. This is a very current and trending topic and published works of the last 5 years should form the basis of the references used.
Response: The authors understand the reviewer's concern regarding this aspect. However, we justify the use of some older references due to their citation in the Material item, aiming to be faithful to the creators of the method, and in some cases, to the creators of some specific knowledge
General Comments
- There is need the authors to really underscore the mechanism behind Cedrela odoratanot distinguishing between nitrate and ammonium in the N absorption process because this study shows that the proportions between these forms of N did not affect its photosynthetic rate, nutrient accumulation or growth.
- I suggest that authors do a thorough language editing of the manuscript
Response: Corrected
- I discourage authors from using personal pronouns throughout the manuscript
Response: Adjusted
Reviewer 2 Report
Comments and Suggestions for Authors
The researchers conducted a study to examine how the proportions of nitrate and ammonium in fertilizers affect the growth, nutritional status, and photosynthetic parameters of Cedrela odorata seedlings for 150 days. The ratios of nitrate to ammonium used in the experiment were 100/0, 80/20, 60/40, 40/60, 20/80, and 0/100, respectively. The study also included a control group that received no nitrogen supply.
The paper is intriguing as it examines how different nitrogen sources affect a certain culture's growth and development. However, the paper falls short in drawing comparisons with other cultures. To address this, I recommend that the authors either base their work on similar research and compare the results, or conduct additional experiments to expand the scope of their findings.
While the analysis results are promising, I must note that the comparison with other references that address similar research is inadequate.
The authors should comprehensively analyze their results by comparing them to similar studies in the existing literature.
Comments on the Quality of English LanguageMinor editing of English language required
Author Response
REVIEWER 2
Comments and Suggestions for Authors
The researchers conducted a study to examine how the proportions of nitrate and ammonium in fertilizers affect the growth, nutritional status, and photosynthetic parameters of Cedrela odorata seedlings for 150 days. The ratios of nitrate to ammonium used in the experiment were 100/0, 80/20, 60/40, 40/60, 20/80, and 0/100, respectively. The study also included a control group that received no nitrogen supply.
The paper is intriguing as it examines how different nitrogen sources affect a certain culture's growth and development. However, the paper falls short in drawing comparisons with other cultures. To address this, I recommend that the authors either base their work on similar research and compare the results, or conduct additional experiments to expand the scope of their findings.
While the analysis results are promising, I must note that the comparison with other references that address similar research is inadequate.
The authors should comprehensively analyze their results by comparing them to similar studies in the existing literature.
Response:
Dear Reviewer,
The authors partially agree with their observations and concerns regarding this aspect. In fact, this article represents a contribution to a better understanding of the nitrogen nutrition of tree species, in the initial growth phase, in particular, on nitrogen forms (ammonium and nitrate). The literature has few studies on mineral nutrition of tree species, mainly on nitrogen nutrition of native species, a fact that sometimes forces us to use other species as references in some discussions. In any case, we strive to, whenever possible, mention articles that mention tree species, native or not, as per the list below.
[6] Cedrela fissilis Vell
[7] Trees species (Jiang et al. 2018)
[8] Atlantic forest tree seedlings
[11] Castro-Rodríguez et a. 2017
[18] Typical steppe ecosystem
[23] Citrus seedlings
[46] Figueiredo et al. 2014
[47] Marenco et al. 2014
[49] Gonçalves et al. 2012
[50] Moretti et a. 2011
[51] Eucalyptus urophylla
[57] Olea europaea
[58] Pinus radiata
